# ADAPTIVE NEURONAL DYNAMICS DRIVE ACCURATE AND EFFICIENT SPIKING NEURAL NETWORKS

## ABSTRACT

Neuronal dynamics are fundamental to the temporal processing capabilities of spiking neural networks (SNNs), directly influencing their responsiveness and integration of dynamic input features. While most existing advances in SNNs focus on architectural optimizations or learning algorithms, the potential of adaptive neuronal dynamics remains underexplored. In this work, we pioneer a new direction by proposing a novel neuron model that achieves adaptive neuronal dynamics through a combined mechanism of learnable and dynamically modulated membrane time constant. This design enables each neuron to regulate its integration behavior in response to stimulus intensity and firing history, achieving short-term adaptive responses while maintaining long-term stability. Importantly, our approach maintains computational simplicity throughout the implementation. Experiments on multiple benchmarks, including physiological signals (DEAP) and neuromorphic vision datasets (DVS-Gesture, CIFAR10-DVS), demonstrate superior performance in both classification accuracy and robustness. In particular, our model achieves state-of-the-art accuracy of 90.11% and 92.57% on the DEAP valence and arousal classification tasks, representing improvements of +9.08% and +9.97% over the baseline. Furthermore, the proposed neuron demonstrates strong generalization across various network architectures, confirming its generality as a fundamental component for SNN-based systems. Additional energy consumption analysis confirms the high efficiency of our method. These results establish adaptive neuronal dynamics as a new pathway for developing effective and efficient neuromorphic systems.

## 1 INTRODUCTION

Spiking neural networks (SNNs) have attracted growing attention due to their event-driven operation and temporal coding capabilities (Li et al., 2021a; Zhou et al., 2021), offering inherent advantages in energy-efficient computation and effective processing of dynamic, time-varying signals (Maass, 1997; Kasabov et al., 2013; Yu et al., 2020; Sun et al., 2022). These characteristics make SNNs particularly suitable for applications requiring rapid response and low power consumption, especially when integrated with neuromorphic hardware (Farsa et al., 2019; Wu et al., 2022).

Neuronal dynamics, describing the time-varying electrical behavior and state evolution of spiking neurons, form the computational foundation of spiking neural networks and play an essential role in determining their information processing capabilities. These dynamics are profoundly influenced by intrinsic cellular properties, among which the membrane time constant $\tau$ plays a particularly pivotal role (Abbott & Dayan, 2001; Gerstner & Kistler, 2002; Izhikevich, 2007). The choice of $\tau$ therefore critically influences the temporal dynamics, responsiveness, and information processing capabilities of SNNs.

Despite this importance, a common practice in most SNNs is to treat the membrane time constant as a fixed hyperparameter or, in limited cases, a globally learnable scalar (Wu et al., 2018; Fang et al., 2021; Perez-Nieves et al., 2021). This static setup significantly constrains the adaptability of neurons, limiting their ability to dynamically modulate membrane potential leakage in response to varying temporal patterns, input strengths, or spiking history. As a result, this limits their ability to process a wide range of temporal data effectively. In contrast, biological neurons exhibit dynamic time constants that continuously adapt to their input conditions and firing activities (Koch et al.,

1996). Experimental studies have shown that strong synaptic drive or high firing rates can accelerate membrane decay (Kuhn et al., 2004; Ha & Cheong, 2017), while weak inputs or sparse firing result in longer integration times, thereby ensuring information processing across multiple timescales (Gerstner & Kistler, 2002). This biological evidence suggests that dynamic adjustment of the time constant is a crucial mechanism for temporal adaptability.

Early dynamic time constant mechanisms can be traced back to biophysical neuron models, such as Hodgkin–Huxley formulations and conductance-based neuron model, where input-dependent conductance changes naturally yield adaptive membrane time constants (Hodgkin & Huxley, 1952; Destexhe, 1997; Gerstner & Kistler, 2002). More recently, the Liquid Time-Constant (LTC) network introduced parameterized recurrent functions to compute time constants dynamically, enabling neurons to flexibly adjust their temporal integration (Hasani et al., 2021; Yin et al., 2023). Following this concept, the Brain-inspired Adaptive Leaky Integrate-and-Fire (BA-LIF) model was proposed, in which the time constant is modulated through a sequence of continuous-valued operations, enabling input-driven adaptation in spiking neurons (Zhang et al., 2025a). Despite their biological plausibility or functional adaptability, these approaches share a common limitation: the dynamic mechanism introduce substantial computational overhead, which increase energy cost and hinder deployment in large-scale neuromorphic systems. Moreover, these methods primarily focus on input-driven instantaneous adaptation, overlooking the potential of learnable long-term plasticity in time constants that could be optimized globally for specific tasks. This limitation motivates our design of a lightweight, dynamic and learnable temporal mechanism that achieves high adaptability while maintaining low computational overhead.

To overcome the above limitations, inspired by biological neurons that exhibit both long-term plasticity (Rimmer & Harper, 2006; Hong et al., 2016) and short-term dynamic modulation of their membrane time constant (Kuhn et al., 2004; Ha & Cheong, 2017), we propose a novel Leaky Integrate-and-Fire neuron model with Adaptive Neuronal Dynamics, named AND-LIF, that enables both dynamic modulation and learned adaptation of the neuron's temporal characteristic. Each neuron is equipped with a trainable base time constant that captures long-term plasticity, with its value dynamically modulated based on recent inputs and spiking activity. This design enables adaptive neuronal dynamics by allowing neurons to continuously adjust their temporal integration behavior in response to incoming signals and firing history, thereby achieving short-term adaptability while maintaining long-term stability.

Since our proposed AND-LIF neuron is designed to enhance the temporal adaptability of spiking neurons via dynamic $\tau$, it is particularly suited for challenging tasks containing rich temporal dynamics. Therefore, in our experiments we focus on dynamic benchmarks, including electroencephalography (EEG) dataset DEAP (Koelstra et al., 2011) as well as the neuromorphic benchmarks such as DVS-Gesture (Amir et al., 2017) and CIFAR10-DVS (Li et al., 2017). Across these datasets, our method consistently improves classification accuracy and maintains low computational overhead, demonstrating its efficacy in enhancing the overall performance of SNNs with biologically inspired short-term adaptation and long-term stability. The main contributions of this work are threefold:

- We introduce a biologically inspired mechanism for dynamic time constants in spiking neural networks, where the time constant is modulated both by inputs and by the neuron's own spike history, enabling local and activity adaptive temporal processing without introducing additional network branches.

- We combine a learnable, globally optimized base time constant with dynamic regulation based on neuronal activity. This unification allows each neuron to maintain long-term stability through learning while retaining short-term adaptive responses to neuronal activities.

- Our proposed AND-LIF neuron achieves state-of-the-art (SOTA) accuracy of 90.11% and 92.57% respectively on the challenging three-class valence and arousal emotion classification task of the DEAP dataset, while also consistently improving performance across DVS-Gesture and CIFAR10-DVS. More importantly, our energy consumption analysis confirms that adaptive neuronal dynamics can be both effective and efficient.

## 2    RELATED WORK

Unlike conventional artificial neural networks that process information through continuous activations, SNNs rely on sparse, discrete spikes for communication. This event-driven approach enables efficient low-latency inference and reduces power consumption when implemented on neuromorphic hardware (Maass, 1997; Kasabov et al., 2013; Kudithipudi et al., 2025). Among the many neuron models, the Leaky Integrate-and-Fire (LIF) model stands out as a widely adopted abstraction due to its simplicity and biological plausibility (Yu et al., 2020; Rajakumari & Pradhan, 2022; Mishra et al., 2025). The LIF neuron integrates input currents into a membrane potential with an exponential decay governed by the membrane time constant. Once the potential crosses a threshold, a spike is emitted and the potential resets. While effective in capturing basic neural dynamics, the non-differentiable spiking function in LIF neurons poses a fundamental challenge for gradient based learning.

To overcome the non-differentiability of spike generation, surrogate gradient methods were introduced(Neftci et al., 2019; Wu et al., 2018; Zenke & Ganguli, 2018). These methods use a continuous surrogate function to approximate the gradient of the spiking function, thereby enabling backpropagation through time in SNNs. By allowing gradients to propagate through spiking dynamics, these methods not only achieve competitive performance with deep artificial neural networks but also facilitate the optimization of additional parameters within the neuron model itself.

Building on the surrogate gradient framework, researchers proposed the Parametric LIF (PLIF) model (Fang et al., 2021; Perez-Nieves et al., 2021), in which the membrane time constant is parameterized and learned during training. Subsequent works have further extended this idea through methods such as Diet-SNN (Rathi & Roy, 2021), which introduces layer-wise learnable time constants, and DA-LIF (Zhang et al., 2025b), which enables spatio-temporally adaptive time constants. Nevertheless, these learnable based models remain fundamentally static: once trained, the time constant is fixed during inference, limiting their ability to adapt to varying temporal patterns in real time.

Early biologically inspired neuron models, such as the Hodgkin–Huxley model (Hodgkin & Huxley, 1952) and conductance-based model (Destexhe, 1997), realize dynamic membrane time constants through voltage-gated ion channels or synaptic conductances, but require solving nonlinear differential equations or costly conductance-dependent updates, making them computationally expensive. More recent approaches, including LTC networks (Hasani et al., 2021; Yin et al., 2023) and BA-LIF neuron model (Zhang et al., 2025a), generate neuron-specific time constants via additional learnable layers, pooling, concatenation, and nonlinear transformations. While these designs enable short-term input-dependent adaptation, they introduce substantial computational and energy overhead and often lack mechanisms for long-term optimization of the membrane time constant. Across these methods, the common limitation is that dynamic adaptation comes at the cost of high complexity, which hinders efficient deployment in spike-based neuromorphic systems.

## 3    METHOD

In this section, we first revisit the LIF neuron model and its training via surrogate gradient backpropagation, which forms the foundation of modern SNN optimization. Building upon this, we introduce our AND-LIF neuron, designed to enable adaptive neuronal dynamics while achieving short-term adaptive responses and preserving long-term stability.

### 3.1    LEAKY INTEGRATE-AND-FIRE NEURON MODEL

SNNs typically employ the LIF model to describe neuronal dynamics. As shown in Fig. 1(a), the LIF neuron integrates incoming current into its membrane potential, while exhibiting a gradual leak of its membrane potential over time. Typically, the dynamics can be expressed as:

$$S^l(t) = \Theta\left(H^l(t) - V_{\text{th}}\right) = \begin{cases} 1, & H^l(t) \geq V_{\text{th}} \\ 0, & H^l(t) < V_{\text{th}} \end{cases}, \tag{1}$$

$$H^l(t) = U^l(t) + I^l(t), \tag{2}$$

$$U^l(t) = \rho_{\text{m}}\left(H^l(t-1) - S^l(t-1)V_{\text{th}}\right), \tag{3}$$

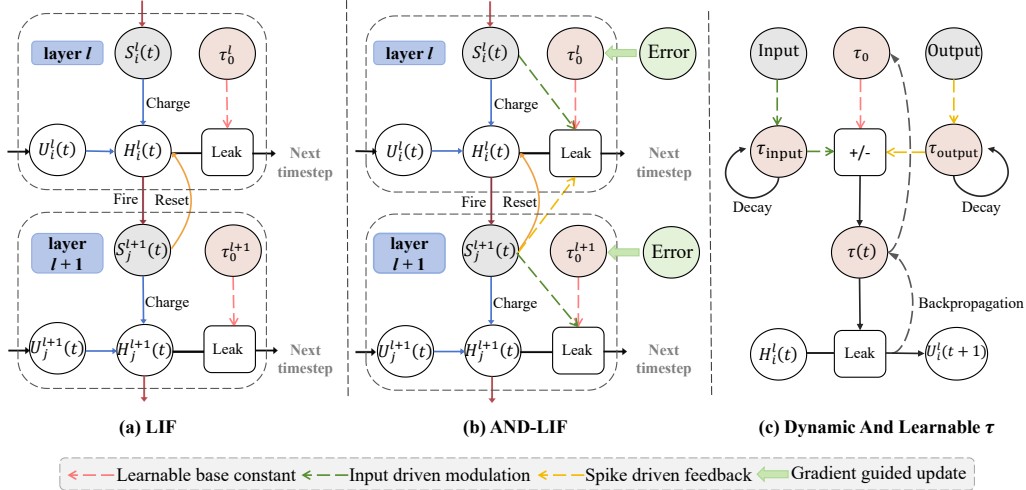

Figure 1: Illustration of the evolution from conventional LIF neurons to our proposed AND-LIF neurons. (a) Standard LIF neuron dynamics, where the membrane potential decays with a fixed time constant. (b) AND-LIF model, in which the time constant is learnable and adaptively modulated by both input and output spike activity. (c) Detailed view of our proposed mechanism, where the membrane time constant $\tau(t)$ is jointly determined by a learnable base constant $\tau_0$, input-driven modulation $\tau_{\text{input}}$, and spike-driven feedback $\tau_{\text{output}}$. Moreover, $\tau_0$ is updated through error backpropagation, enabling an efficient combination of plasticity and adaptability.

$$I^l(t) = f\left(W^l, S^{l-1}(t)\right), \tag{4}$$

where $S^l(t)$ and $H^l(t)$ represent the spike sequence and the membrane potential at time step $t$ for the $l$-th layer, respectively. $U^l(t)$ represents the voltage after membrane leakage and $V_{\text{th}}$ is the threshold that determines whether $H^l(t)$ results in a spike or remains silent. $I^l(t)$ indicates the input at time step $t$ for the $l$-th layer and $f(\cdot)$ is the function operation stands for convolution or fully connected computation. $W^l$ denotes synaptic weight matrix between two adjacent layers and $\rho_{\text{m}}$ is the membrane decay factor, which controls the retention of past membrane potential at each timestep. The decay parameter $\rho_{\text{m}}$ essentially governs the rate at which the membrane potential decays toward the resting state. In practice, this decay factor is intrinsically linked to the membrane time constant $\tau$ through the exponential representation of the temporal decay dynamics:

$$\rho_{\text{m}} = e^{-\frac{\Delta t}{\tau}}, \tag{5}$$

where $\Delta t$ denotes the simulation timestep.

As demonstrated by Eq. 1, the non-differentiable characteristics of spiking activity preclude direct application of backpropagation to SNNs. To address this fundamental limitation, surrogate gradient approaches replace the undefined gradients of spike generation with surrogate gradient functions $u(\cdot)$ during error backpropagation. Specifically, the SNN computes the spike activity via Eq. 1 and its derivative with respect to the membrane potential through Eq. 6:

$$\frac{\partial S^l(t)}{\partial U^l(t)} \approx u'\left(U^l(t), V_{\text{th}}\right), \tag{6}$$

For example, in our experiments on the DVS-Gesture dataset, we follow (Wu et al., 2018; 2022; Ding et al., 2024b) and adopt the rectangular function as the surrogate gradient:

$$u'\left(U^l(t), V_{\text{th}}\right) = \frac{1}{a}\,\text{sign}\left(\left|U^l(t) - V_{\text{th}}\right| < \frac{a}{2}\right), \tag{7}$$

where $a$ serves as the hyperparameter controlling the shape of the rectangular function and we set it to 1 for general simplicity. Through this way, chain rule-based parameter updates can be achieved.

## 3.2 Adaptive Neuronal Dynamic Leaky Integrate-and-Fire neuron model

Biological studies have revealed that neuronal dynamics are influenced by a wide range of factors, encompassing both long-term plasticity and short-term adaptive mechanisms (Koch et al., 1996; Rimmer & Harper, 2006; Hong et al., 2016; Kuhn et al., 2004; Ha & Cheong, 2017). For instance, it has been demonstrated that intense synaptic input reduces the membrane time constant through increased conductance (Kuhn et al., 2004), while sustained spiking activity has been shown to trigger calcium-mediated adaptation that shortens integration windows (Ha & Cheong, 2017). To integrate these adaptive processes into a unified computational framework, we propose a generalized form of the adaptive time constant:

$$\tau(t) = F(\tau_0, I(t), S(t)), \tag{8}$$

where $\tau_0$ represents the base value of the membrane time constant. Based on Eq. 6, $\tau_0$ can be optimized through backpropagation by treating it as a learnable parameter. While such adaptive behavior $F(\cdot)$ could be modeled through complex biophysical equations or additional network layers (Hodgkin & Huxley, 1952; Destexhe, 1997; Hasani et al., 2021; Zhang et al., 2025a), we prioritize computational efficiency and hardware compatibility. Our design captures these reverse relationships in a lightweight manner through a unified modulation function that incorporates both input intensity and output spiking activity. This leads to the formulation presented in Eq. 9, which provides a biologically inspired mechanism for dynamic time constant adaptation. Specifically, in our method, the membrane time constant $\tau(t)$ is dynamically adjusted at each timestep through:

$$\tau(t) = \tau_0 - \tau_{\text{input}}(t) - \tau_{\text{output}}(t), \tag{9}$$

where the terms $\tau_{\text{input}}(t)$ and $\tau_{\text{output}}(t)$ correspond to the adjustments caused by the input and output activities, respectively. The dynamics of these adjustments are described by the following relations:

$$\tau_{\text{input}}(t) = \lambda_i \tau_{\text{input}}(t-1) + \eta_i I(t), \tag{10}$$

$$\tau_{\text{output}}(t) = \lambda_o \tau_{\text{output}}(t-1) + \eta_o S(t-1), \tag{11}$$

$\tau_{\text{input}}(t)$ is influenced by the input $I(t)$ at time $t$, with $\lambda_i$ controlling the persistence of the previous input adjustments, and $\eta_i$ determining the influence of the current input on the adjustment. Similarly, $\tau_{\text{output}}(t)$ is driven by the spike $S(t-1)$ from the previous time step, where $\lambda_o$ regulates the decay of past output-driven adjustments, and $\eta_o$ dictates how much the output spike influences the current adjustment. As shown in Fig. 1(b, c), these parameters work together to enable dynamic modulation of the membrane time constant, allowing the neuron to better respond to changing input stimuli and its own firing behavior. This dynamic and learnable mechanism makes the model more flexible and capable of handling various temporal patterns efficiently.

Based on this, the new membrane decay factor $\rho_{\text{m}}$ and membrane potential update formula $U^l(t)$ can be expressed as follows:

$$\rho_{\text{m}}(t) = e^{-\frac{\Delta t}{\tau(t)}}, \tag{12}$$

$$U^l(t) = \rho_{\text{m}}(t)\left(H^l(t-1) - S^l(t-1)V_{\text{th}}\right), \tag{13}$$

The above equations define the dynamics of our proposed neuron model. Here, the membrane decay factor is dynamically modulated by three components: a learnable baseline, an input-dependent term, and an output-dependent term. This modulation directly governs the subsequent update of the membrane potential. This mechanism allows each neuron to flexibly adjust its temporal integration based on current input, past activity, and its own intrinsic properties, achieving both short-term adaptive responses and long-term stability. As a result, the neuron can integrate information efficiently over varying timescales, enhancing the overall capability of spiking neural networks.

## 4 Experiments

In this section, we evaluate the proposed AND-LIF neuron across a diverse range of dynamic benchmarks, to validate its effectiveness and generalization capability. To reduce randomness, we report the average results of three independent experiments in our experiments. For detailed information regarding the experimental setup, refer to Appendix A.1.

Table 1: Comparative results on DEAP datasets. Valence and Arousal refer to the two primary affective dimensions in emotion recognition. The experiments are conducted on both 2-class and 3-class classification settings.

| Dataset | Method | Class | Valence (%) | Arousal (%) |
|---|---|---|---|---|
| DEAP | Transfer learning (Yan et al., 2022) | 2 | 82.75 | 84.22 |
| | SNN+IIR (Xu et al., 2024) | 2 | 61.15 | 53.86 |
| | EEGNet (Gong et al., 2023) | 2 | 81.14 | 76.82 |
| | SGLNet (Gong et al., 2023) | 2 | 90.01 | 89.41 |
| | SCNN (Islam et al., 2021) | 3 | 70.23 | 70.25 |
| | DH-SNN (Zheng et al., 2024) | 3 | 77.46 | 80.23 |
| | BSNN (Sun et al., 2025) | 3 | 81.03 | 82.60 |
| | **AND-LIF (ours)** | 2 | **92.35** | **94.31** |
| | | 3 | **90.11** | **92.57** |

Table 2: Comparative results on neuromorphic datasets. The symbol of * indicates our implementation results with the corresponding released codes under the same platform to ensure fair comparisons. † denotes that the corresponding baseline methods use additional temporal attention module specifically designed to enhance network performance.

| Dataset | Method | Architecture | T | Accuracy (%) |
|---|---|---|---|---|
| DVS-Gesture | STBP (Wu et al., 2018) | 5Conv,3FC | 10 | 87.50* |
| | BA-LIF (Zhang et al., 2025a) | 5Conv,1FC | 20 | 97.90† |
| | SLTT (Meng et al., 2023) | VGG-11 | 5 | 92.02* |
| | SSNN (Ding et al., 2024b) | VGG-9 | 5 | 90.74 |
| | TRR (Zuo et al., 2024) | VGG-9 | 5 | 91.67 |
| | Spikformer (Zhou et al., 2022) | Spikingformer | 4 | 93.40* |
| | **AND-LIF (ours)** | 5conv,3FC | 10 | 96.18 |
| | | VGG-11 | 5 | 95.14 |
| | | Spikingformer | 4 | 94.79 |
| | | Spikingformer | 16 | **98.61** |
| CIFAR10-DVS | PLIF (Fang et al., 2021) | 4Conv,2FC | 10 | 69.15* |
| | SLTT (Meng et al., 2023) | VGG-11 | 10 | 76.40* |
| | BA-LIF (Zhang et al., 2025a) | VGG-11 | 10 | 77.70† |
| | TRR (Zuo et al., 2024) | Spikingformer | 5 | 75.55 |
| | Spikformer (Zhou et al., 2022) | Spikingformer | 16 | 79.40* |
| | **AND-LIF (ours)** | 4Conv,2FC | 10 | 69.79 |
| | | VGG-11 | 10 | 77.20 |
| | | Spikingformer | 4 | 77.70 |
| | | Spikingformer | 16 | **80.41** |

## 4.1 COMPARISON WITH EXISTING METHODS

**DEAP Dataset.** In Tab. 1, we present a comparison of our method against existing SNN-based direct training approaches on the DEAP dataset. We follow the pre-possessing method used by (Zheng et al., 2024). The first 3s data of each trial is used for producing the average 1s baseline signal by averaging the 3s data per second. The following 60s data was normalized by subtracting the average 1s baseline signal every second and then divided into 20 segments of 3s for each. The table

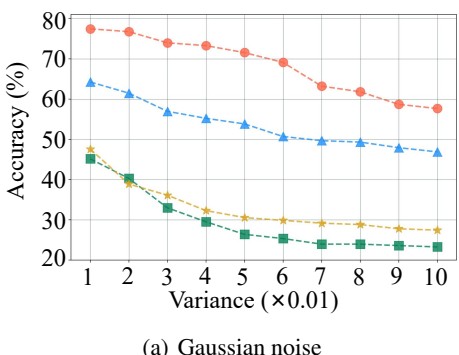 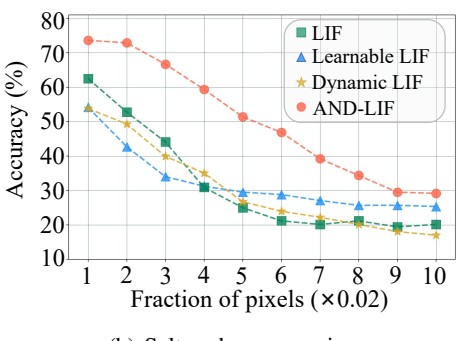

(a) Gaussian noise

(b) Salt-and-pepper noise

Figure 2: Performance on DVS-Gesture under varying levels of (a) Gaussian noise, where the noise level reflects the variance of the zero-mean noise added to each pixel, and (b) salt-and-pepper noise, where the level indicates the fraction of pixels randomly replaced by either minimum or maximum intensity values. Detailed settings can be found in the Appendix A.1.3.

reports classification accuracies for both the Valence and Arousal dimensions, which correspond to the emotional positivity/negativity and intensity, respectively. As shown in Tab. 1, our model with an architecture of two fully-connected layers achieves SOTA accuracy, particularly on the challenging 3-class tasks, reaching 90.11% for Valence and 92.57% for Arousal, representing remarkable improvements of +9.08% and +9.97% over the baseline. These results demonstrate the effectiveness of our approach in capturing complex affective patterns compared to prior SNN models.

**Neuromorphic Dataset.** We evaluate our AND-LIF neurons on three commonly used and effective SNN architectures: feedforward SCNN (Wu et al., 2018; Fang et al., 2021), VGG (Meng et al., 2023; Zhang et al., 2025a) and Spikingformer (Zhou et al., 2022). These architectures provide representative benchmarks for assessing the impact of our dynamic and learnable mechanism. Within each architecture, our AND-LIF is compared against standard LIF neurons and existing neurons incorporating time constant adaptation mechanisms. As shown in Tab. 2, on both the DVS-Gesture and CIFAR10-DVS datasets, our AND-LIF consistently improves classification accuracy across various network configurations: it achieves an improvement of +9.68% with a feedforward SCNN architecture at T=10 and +3.12% with VGG-11 at T=5 on DVS-Gesture, while yielding a gain of +2.25% using Spikingformer at T=4 on CIFAR10-DVS. These consistent enhancements demonstrate the generality and effectiveness of AND-LIF across different network designs. Overall, these results confirm that our AND-LIF delivers consistent performance improvements across diverse architectural contexts, highlighting its general applicability as a replacement for standard LIF neurons.

## 4.2 ROBUSTNESS STUDY

To evaluate the robustness of our method against input perturbations, we introduce two types of noise widely adopted in robustness assessments across domains, including spiking neural networks (Li et al., 2020; Wu et al., 2022; Ding et al., 2024a): Gaussian noise and salt-and-pepper noise. Gaussian noise simulates continuous analog disturbances, while salt-and-pepper noise randomly corrupts a subset of pixels by setting them to either minimum or maximum values. Based on this motivation, we conduct robustness experiments on the DVS-Gesture dataset by imposing these noise types at various levels into the event streams, thereby examining the model's stability under both smooth and extreme corruptions.

Specifically, we subject the input events to Gaussian noise and salt-and-pepper noise, and compare the performance across four neuronal variants: standard LIF, LIF with learnable $\tau$, LIF with dynamic $\tau$, and our proposed AND-LIF. The accuracy degradation curves under increasing noise levels in Fig. 2 illustrates that AND-LIF maintains superior performance compared to the other variants. These results indicate that the combination of dynamic and learnable mechanisms not only elevates accuracy but also enhances resilience against input noise.

Table 3: Comparison of energy consumption per timestep on the DEAP dataset. The notation 'M' denotes that the counts of addition and multiplication operations are expressed in millions.

| Model | Firing rates (%) | ADD/Multi(M) | Power(mJ)/ Accuracy (%) |
|---|---|---|---|
| LIF(Sun et al., 2025) | 46.68 | 1.049/ - | 0.441/ 79.06 |
| LTC-LIF (Hasani et al., 2021) | 25.42 | 1.573/ 0.524 | 0.852/ 45.86 |
| BA-LIF (Zhang et al., 2025a) | 48.56 | 2.097/ 1.049 | 2.801/ 86.88 |
| **AND-LIF (ours)** | 12.79 | 1.051/ 0.002 | **0.122/ 90.11** |

Table 4: Ablation studies of learnable and dynamic $\tau$ in our method.

| Dataset | Model | Learnable | Dynamic | Accuracy (%) |
|---|---|---|---|---|
| DEAP | LIF | × | × | 79.06 |
| | LIF with learnable $\tau$ | ✓ | × | 82.18 |
| | LIF with dynamic $\tau$ | × | ✓ | 84.14 |
| | **AND-LIF** | ✓ | ✓ | **90.11** |
| DVS-Gesture | LIF | × | × | 87.50 |
| | LIF with learnable $\tau$ | ✓ | × | 95.14 |
| | LIF with dynamic $\tau$ | × | ✓ | 95.49 |
| | **AND-LIF** | ✓ | ✓ | **96.18** |

## 4.3 ENERGY CONSUMPTION ANALYSIS

The quest for low power drives the development of spiking neural networks, making energy efficiency a central metric in neuromorphic computing. Following (Li et al., 2021b; Rathi & Roy, 2021; Zhang et al., 2025a), we calculate the firing rates, multiply operations, accumulate operations and energy cost. To eliminate the influence of complex network architectures on energy estimation, we conduct the energy consumption analysis on the DEAP dataset using a consistent two-layer fully connected network as an example. This setup ensures that the results primarily reflect the contribution of the dynamic $\tau$ mechanism, rather than being biased by architectural complexity.

Since the introduction of dynamic $\tau$ mechanisms inevitably introduces additional computational overhead, we specifically compare among models that incorporate adaptive time constants. Therefore, we analyze the theoretical energy consumption of different neuron models, focusing on LIF (Sun et al., 2025), LTC-LIF (Hasani et al., 2021), BA-LIF (Zhang et al., 2025a), and our AND-LIF. Specifically, the energy is measured in 45nm CMOS technology. The multiply operation costs 3.7pJ energy and the accumulation costs 0.9pJ energy (Horowitz, 2014). As shown in Tab. 3, our method significantly reduces computational operations compared to other dynamic approaches. Notably, although AND-LIF introduces additional computational operations, it substantially reduces the firing rate, resulting in an overall energy consumption of only 0.122 mJ while achieving 90.11% accuracy—a level that approaches one-quarter of the energy cost of the standard LIF model. This highlights the superiority of our approach in balancing computational cost and spiking efficiency.

## 4.4 ABLATION STUDY

**Contributions of Dynamic and Learnable Mechanism.** To disentangle the contributions of the dynamic and learnable components in our AND-LIF neuron, we conduct ablation experiments on both the DEAP and DVS-Gesture datasets. The results in Tab. 4 show that both learnability and dynamics contribute positively to performance. Introducing either mechanism leads to a clear improvement over the baseline LIF. Furthermore, combining both leads to a further enhancement. In particular, AND-LIF achieves an accuracy gain of 11.05% on DEAP and 8.68% on DVS-Gesture over the standard LIF, validating our design choice of integrating dynamic flexibility with learnable adaptability into a unified neuron model.

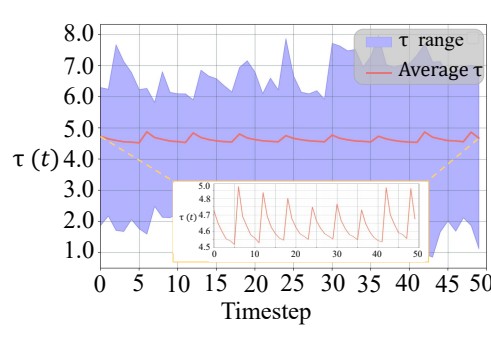

(a) Dynamic evolution

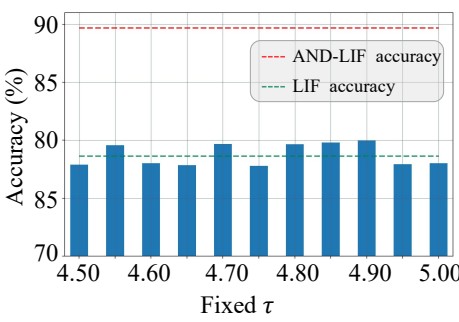

(b) Accuracy with fixed $\tau$

Figure 3: Ablation study on the dynamic $\tau$ mechanism. (a) Temporal evolution of $\tau$ in the AND-LIF neuron during inference. The shaded area indicates the fluctuation range caused by input- and output-driven modulation while the solid line represents the mean value. The inset panel represents an enlarged visualization. (b) Accuracy comparison when $\tau$ is fixed at values within the observed dynamic range after training, showing a significant performance drop once dynamic updates are removed.

**Contributions of Dynamic Mechanism to Adaptability.** To further evaluate the contribution of the dynamic membrane time constant, we visualize its temporal evolution during inference on the DEAP dataset in Fig. 3(a). The figure shows that $\tau$ fluctuates within a certain range over time, reflecting input- and output-driven modulations. To further assess the importance of this dynamic behavior, we conduct an experiment where $\tau$ is fixed within this observed average range after training in Fig. 3(b). We find that disabling the dynamic updates leads to a substantial drop in accuracy, demonstrating that the temporal adaptation of $\tau$ is crucial for model performance. Notably, after removing the dynamic component, the accuracy decreases to a level close to that of a standard LIF neuron, confirming that the observed gains originate primarily from the dynamic mechanism rather than other factors.

**Contributions of Input and Output Components.** To understand the relative importance of input-driven and output-driven contributions in $\tau$ updates, we conduct experiments selectively enabling or disabling each component on the DEAP dataset. The results, summarized in Tab. 5, indicate that

Table 5: Ablation of input- and output-driven components.

| Model | Firing rates (%) | Accuracy (%) |
|---|---|---|
| Original Learnable $\tau_0$ | 34.28 | 82.18 |
| $+\tau_{\text{output}}$ | 33.61 | 83.30 |
| $+\tau_{\text{input}}$ | 13.41 | 86.76 |
| $+\tau_{\text{input}}$ & $+\tau_{\text{output}}$ | **12.79** | **90.11** |

both components not only improve accuracy but also reduce firing rates, thereby significantly enhancing network efficiency. This suggests that the neuron benefits from adapting its membrane time constant based on both incoming currents and previous spiking activity.

## 5 CONCLUSION

Inspired by biological mechanisms, we propose the AND-LIF neuron model to achieve adaptive neuronal dynamics—a pioneering exploration into the advantages of brain-inspired adaptability at the neuronal level. While most existing advances in spiking neural networks focus on architectural optimizations or learning algorithms, this work establishes a new direction by demonstrating how adaptive neuronal dynamics can significantly enhance network capability. This model enables neurons to maintain long-term stability and short-term adaptability through a combined design of both learnable and dynamically modulated time constants. Extensive experiments demonstrate that the proposed neuron not only improves SNN performance and robustness but also significantly enhances operational efficiency. Additional evaluations across various network architectures confirm the broad applicability and generalizability of our method. These results highlight the AND-LIF neuron as a lightweight yet powerful component for enhancing temporal processing in spiking neural networks, opening a new pathway for developing effective and efficient neuromorphic systems.

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

## A  APPENDIX

### A.1  EXPERIMENTAL DETAILS

#### A.1.1  DATASETS

In this paper, we perform experiments on the EEG dataset DEAP and the neuromorphic datasets CIFAR10-DVS, DVS-Gesture.

DEAP (Koelstra et al., 2011): A multi-modal dataset for the analysis of human affective states. It contains electroencephalography and other physiological recordings from 32 participants, each watching 40 one-minute-long music videos. The dataset is labeled with ratings for arousal, valence, dominance, and liking, making it a standard benchmark for emotion recognition tasks using physiological signals.

DVS-Gesture (Amir et al., 2017): A neuromorphic benchmark dataset for gesture recognition, recorded using a dynamic vision sensor. It contains 1,464 instances of 11 different hand and arm gestures performed by 29 subjects under three different lighting conditions. The event-based nature of the data makes it ideal for testing the temporal dynamics and efficiency of spiking neural networks.

CIFAR10-DVS (Li et al., 2017): A neuromorphic version of the popular CIFAR-10 dataset, converted by displaying the original images on a monitor and recording the output with a dynamic vision sensor. It contains 10,000 event streams spanning 10 object classes, presenting a challenging benchmark for object recognition with event-based cameras that requires handling sparse, asynchronous inputs.

#### A.1.2  TRAINING SETUP

Experiments are implemented using PyTorch (version 2.0.1) with CUDA 11.8 and conducted on a server equipped with NVIDIA RTX 4090 GPUs. In the experiment section, the details of the simulation time steps, training epochs, batch size, and some neuronal hyperparameters are summarized in Tab. 6. For the DEAP dataset, we adopt a two-layer fully connected network structure with

Table 6: Parameter settings on different learning tasks.

| Parameters | Descriptions | DEAP | DVS-Gesture | CIFAR10-DVS |
|---|---|---|---|---|
| Batch size | - | 200 | - | - |
| $T$ | Time steps | 6 | - | - |
| $\lambda_i, \lambda_o$ | Decay factor | 0.1 | 0.35 | 0.2 |
| $\eta_i, \eta_0$ | Activity impact factor | 0.01 | 0.1 | 0.05 |
| $V_{\text{th}}$ | Threshold | 1.0 | - | - |
| $N$ | Training epochs | 100 | - | - |

dimensions [512, 3/2], and the detailed hyperparameter settings are provided in Tab. 6. For the neuromorphic vision datasets DVS-Gesture and CIFAR10-DVS, as experiments were conducted across different network architectures, parameters such as the number of time steps and the firing threshold are not fixed, but instead follow the configurations used in previous works (Wu et al., 2018; Fang et al., 2021; Meng et al., 2023; Zhou et al., 2022). The only modification in these experiments is the replacement of the original LIF or PLIF neurons in the reference architectures with our proposed AND-LIF neurons for a fair comparison.

#### A.1.3  ROBUSTNESS EXPERIMENTS

We evaluate the robustness of our model under input perturbations. This is critical for neuromorphic systems that operate in noisy environments, such as low-light settings or high-speed motion scenarios (Hendy & Merkel, 2022). We apply two widely used noise types: Gaussian noise and salt-and-pepper noise, both commonly used in robustness testing of SNNs (Li et al., 2020; Wu et al., 2022; Ding et al., 2024a).

**Gaussian noise:** We perturb the input by adding zero-mean Gaussian noise sampled from $\mathcal{N}(0, \sigma^2)$ to each pixel of the image. The standard deviation is controlled via the variance parameter $\sigma^2$. We define ten increasing noise levels as follows:

$$\sigma^2 \in \{0.01,\ 0.02,\ 0.03,\ 0.04,\ 0.05,\ 0.06,\ 0.07,\ 0.08,\ 0.09,\ 0.10\}.$$

Each noise level is applied independently to the test set, and model performance is reported as classification accuracy. This simulates gradually deteriorating sensor quality or transmission interference.

**Salt-and-pepper noise:** This noise simulates dead pixels and random corruption by flipping a proportion of image pixels to 0 (black) or 1 (white). For each level, we randomly select a fraction $a$ of pixels in the image and assign them new values drawn from a binary distribution. The tested levels are:

$$a \in \{0.02,\ 0.04,\ 0.06,\ 0.08,\ 0.10,\ 0.12,\ 0.14,\ 0.16,\ 0.18,\ 0.20\}.$$

This experiment tests the model's ability to recognize structure in partially corrupted input and maintain stability under non-Gaussian perturbations.

### A.2 ABLATION STUDY ON $\lambda$ AND $\eta$ COEFFICIENTS

The selection of hyperparameters plays a critical role in overall network performance. To evaluate the impact of the key hyperparameters $\lambda$ and $\eta$ introduced in our AND-LIF model, we conduct a hyperparameter search on the DEAP dataset (three-class valence emotion classification task). We begin with a coarse-grained evaluation by setting both $\lambda$ and $\eta$ to values in $\{0.001, 0.01, 0.1, 0.2\}$. The corresponding accuracies—80.89%, 88.55%, 86.33%, and 79.88%—clearly indicate that very small (e.g., 0.001) or very large (e.g., 0.2) values significantly degrade performance, underscoring the importance of appropriate hyperparameter balancing. Based on these observations, we perform a finer-grained search within $\lambda$ in $\{0.01, 0.05, 0.1, 0.15\}$ and $\eta$ in $\{0.005, 0.01, 0.05, 0.1, 0.15\}$. As summarized in Tab. 7, the optimal performance is achieved at $\lambda$=0.1 and $\eta$=0.01, yielding 90.11% accuracy on the DEAP dataset. These results confirm that identifying suitable values for $\lambda$ and $\eta$ leads to substantially improved performance of the proposed neuron model.

Table 7: Accuracy under different settings.

| $\lambda$ \ $\eta$ | 0.005 | 0.01 | 0.05 | 0.1 | 0.15 |
|---|---|---|---|---|---|
| 0.01 | 88.71 | 88.55 | 86.72 | 87.65 | 86.52 |
| 0.05 | 87.81 | 88.98 | 88.55 | 87.34 | 87.77 |
| 0.1 | 88.77 | **90.11** | 87.46 | 86.33 | 86.41 |
| 0.15 | 87.34 | 88.86 | 87.42 | 85.62 | 83.71 |

### A.3 USAGE OF LARGE LANGUAGE MODELS

In this study, large language models were employed solely to assist in polishing wording and grammatical structures during the writing process. All scientific content, including theoretical development, methodological design, experimental implementation, data analysis, and conclusions, remains entirely the work of the human authors. The use of AI was strictly limited to linguistic refinement and did not involve any contribution to the intellectual or creative substance of the research.

