# OpenReview forum: "Adaptive Neuronal Dynamics Drive Accurate and Efficient Spiking Neural Networks"
_ICLR.cc/2026/Conference — ICLR 2026 Conference Withdrawn Submission_

### Official Review · Reviewer_77oK · 2025-10-21

**Soundness:** 3
**Presentation:** 3
**Contribution:** 2
**Rating:** 2
**Confidence:** 4

**Summary:**

The parameterization of spiking neurons has a large impact on SNN performance and has been intensively studied. This paper proposes the AND-LIF model, whose membrane time constant τ is decomposed into three terms: a fixed base value, an input-dependent term, and an output-dependent term. The base value is trained by gradient descent, while the input- and output-related terms are updated as moving averages of the input current and output spikes, respectively; the coefficients in these recursions are also learned by back-propagation. The model is evaluated on one EEG classification task and two neuromorphic vision classification tasks.

**Strengths:**

- The motivation is sound: AND-LIF adapts τ on-the-fly to both input and output, which in practice usually enlarges the neuron's memory horizon.
- Compared with the standard LIF neuron, AND-LIF introduces only negligible extra computation.

**Weaknesses:**

The benchmark suite is not sufficiently representative. DEAP has rarely been used in previous SNN works, so well-established baselines are lacking; DVS-Gesture is small, and at >97 % accuracy the difference between methods amounts to only a few correctly classified samples; CIFAR10-DVS is the only large and challenging dataset with extensive baselines. The current results do not conclusively demonstrate the effectiveness of the proposed method.

**Questions:**

How much slower is AND-LIF than LIF in actual training and inference time?

---

### Official Review · Reviewer_rjCE · 2025-10-29

**Soundness:** 2
**Presentation:** 3
**Contribution:** 1
**Rating:** 2
**Confidence:** 5

**Summary:**

The authors use spiking neural networks for image and neuromorphic dataset classification. They propose a new spiking neuron model (the AND-LIF): essentially a leaky integrate-and-fire (LIF) neuron whose time constant is dynamic, because it is input and output-dependent.

**Strengths:**

As far as I know, this neuron model is new.

**Weaknesses:**

* The ablation study (Table 4) is not convincing, because the accuracy of the LIF on DVS-Gesture (I'm not familiar with DEAP, the other dataset they use) is very poor. The AND-LIF does much better, but of course, it's much easier to improve the accuracy on a poor architecture. So I suggest the authors use an architecture that reaches, or approaches, the SOTA with LIF neurons. If the  AND-LIF does even better, then it would improve the SOTA, making the paper much more interesting (more on this below).

* Contrary to what the authors say and the tables suggest, the authors do not reach the SOTA on CIFAR10-DVS and do not improve it on DVS-Gesture (again, I'm not familiar with DEAP, the other dataset they use):
1) On CIFAR10-DVS, several papers (not cited) reached 83%+ (see http://arxiv.org/abs/2304.12760 and refs therein), which is much better than this paper
2) On DVS-Gesture, several papers (not cited) reached 98%+ (see http://arxiv.org/abs/2502.10422 and refs therein), which is similar to this paper

So again, I suggest the authors start from an architecture that reaches the SOTA with LIF, and then replace LIF with AND-LIF, and see if this allows improving the SOTA.

MINOR POINTS

* Fig 1: I think S_j^{l+1} belongs to layer l. Thus, it should be in the upper box, and called S_j^l

* Eq. 3: the authors may want to say that they used a so-called "soft reset".

**Questions:**

Eq 10 and 11: the authors use two parameters lambda and eta, but I think one would do. I think they could use a convex combination with lambda and (1-lambda), right?

---

### Official Review · Reviewer_RK65 · 2025-10-30

**Soundness:** 3
**Presentation:** 2
**Contribution:** 2
**Rating:** 4
**Confidence:** 4

**Summary:**

The paper proposes AND-LIF, a novel spiking neuron model that combines learnable and dynamically modulated membrane time constants. The neuron adaptively regulates its temporal integration behavior based on both input strength and spike history, achieving short-term adaptability while maintaining long-term stability.

**Strengths:**

This paper grounds its design in well-known neurophysiological phenomena (e.g., Hodgkin–Huxley conductance adaptation, short-term plasticity), bridging biological neuron dynamics and computational models.
The proposed design integrates a trainable base time constant (long-term plasticity) with input- and spike-driven dynamic modulation (short-term adaptability).

**Weaknesses:**

1. The proposed adaptive time constant mechanism builds on well-established ideas from LTC networks and BA-LIF. The paper’s novelty lies more in combining learnable and dynamic components rather than introducing a fundamentally new formulation of temporal adaptation.
2. The contributions of learnable vs. dynamic modulation are not thoroughly separated. It remains unclear how much each component contributes to the final efficiency gains.
3. Although DVS-Gesture and DEAP are appropriate dynamic datasets, the paper does not test on frame-based temporal tasks (e.g., video or event-to-frame hybrid benchmarks). This limits the understanding of how AND-LIF scales to complex spatio-temporal modeling.
4. This work focuses at the neuron level, but does not explore how adaptive dynamics could interact with higher-level architectural components.
5. Without hardware-level validation, the energy benefit remains somewhat speculative.

**Questions:**

see weakness

---

### Note · Authors · 2025-11-21

I have read and agree with the venue's withdrawal policy on behalf of myself and my co-authors.